# Genetic Inheritance Models of Non-Syndromic Cleft Lip with or without Palate: From Monogenic to Polygenic

**DOI:** 10.3390/genes14101859

**Published:** 2023-09-24

**Authors:** Xi Cheng, Fengzhou Du, Xiao Long, Jiuzuo Huang

**Affiliations:** 1Peking Union Medical College Hospital, Chinese Academy of Medical Sciences & Peking Union Medical College, Beijing 100730, China; cheng-x17@mails.tsinghua.edu.cn (X.C.); fengzhoudu@163.com (F.D.); pumclongxiao@126.com (X.L.); 2Department of Plastic Surgery, Peking Union Medical College Hospital, Beijing 100730, China

**Keywords:** non-syndromic cleft lip with or without palate (NSCL/P), genetic etiology, inheritance model, polygenic risk score (PRS)

## Abstract

Non-syndromic cleft lip with or without palate (NSCL/P) is a prevalent birth defect that affects 1/500–1/1400 live births globally. The genetic basis of NSCL/P is intricate and involves both genetic and environmental factors. In the past few years, various genetic inheritance models have been proposed to elucidate the underlying mechanisms of NSCL/P. These models range from simple monogenic inheritance to more complex polygenic inheritance. Here, we present a comprehensive overview of the genetic inheritance model of NSCL/P exemplified by representative genes and regions from both monogenic and polygenic perspectives. We also summarize existing association studies and corresponding loci of NSCL/P within the Chinese population and highlight the potential of utilizing polygenic risk scores for risk stratification of NSCL/P. The potential application of polygenic models offers promising avenues for improved risk assessment and personalized approaches in the prevention and management of NSCL/P individuals.

## 1. Background

As the most common craniofacial anomalies worldwide, orofacial clefts (OFCs) present in 1/700 live births worldwide [1]. The phenotype spectrum associated with OFCs is highly variable and involves only lip (cleft lip, CL), only palate (cleft palate, CP), and the combination thereof: cleft lip with or without palate (CL/P). Each phenotype could be further categorized as unilateral/bilateral, and complete/incomplete [2]. Orofacial clefts could substantially affect a child’s health and overall well-being, including difficulties with feeding, speech, hearing, and social interaction. These anomalies not only affect patients’ aesthetics but also could result in significant psychological and financial burdens to the family in some severe cases.

The embryogenesis of the lip and palate has been extensively and elaborately reviewed [2,3,4,5]. During the development of the upper lip (between the 4th and 8th weeks of gestation), the mesenchymal cell proliferation and tissue augmentation give rise to the bilateral medial and lateral nasal prominences encompassing the nasal placode. The medial nasal prominences and the in-between intermaxillary segment continue to grow and merge into the philtrum at the 5th week of gestation [4]. Then, the bilateral maxillary prominences progress medially and coalesce with the philtrum, producing the continuous edge of the upper lip. Palatogenesis commences from the 6th week to the 12th week. Following the fusion of the philtrum and maxillary prominences, the tissue underlying the nasal pits proliferates, giving rise to the median palatine process (primary palate), which demarcates nostrils and upper lip. Subsequently, paired lateral palatine processes originate from the maxillary processes and extend horizontally toward the midline beneath the nasal septum during the 8th to 9th week of gestation [5], ultimately forming the secondary palate [4]. The embryogenesis of the upper lip and palate encompasses overlapping temporal phases and distinct embryonic origins. Disruptions in these processes can lead to varying degrees of severity and phenotypic presentation. A cleft lip arises from the failed integration of the philtrum and maxillary prominences on one or both sides, while a cleft palate could be caused by the failed merging of paired lateral palatine processes. The study of embryogenesis not only offers insights into the fundamental mechanisms governing craniofacial structure development but also enables researchers to identify potential genetic, epigenetic, and environmental factors that may play a role in the occurrence of orofacial clefts.

Depending on the involvement of other affected systems, CL/P can be categorized into syndromic CL/P (SCL/P) and non-syndromic CL/P (NSCL/P) (Figure 1). SCL/P often occurs as a craniofacial phenotype within syndromes that affect multiple systems, e.g., neurologic, cardiovascular, and/or skeletal systems. SCL/P is typically caused by genetic mutations or abnormalities that affect multiple aspects of development. Therefore, individuals with syndromic CL/P may exhibit a range of additional health issues or physical characteristics beyond the cleft itself. In contrast, NSCL/P exhibits the isolated presentation of clefts at the lip and/or palate without the involvement of other systems. It represents the predominant type of CL/P, constituting about 70% of OFC cases [6], with a prevalence of 1/500–1/1400 among ethnicities [7]. The prevalence of NSCL/P is highest in the White maternal race, followed by the Asian and Hispanic maternal race, and lowest in the Black maternal race [8,9]. Additionally, it tends to be more frequently diagnosed in males [10].The etiology of NSCL/P has long been an intense investigation topic within the realm of craniofacial research. Historically, the search for NSCL/P genetic underpinnings has evolved from monogenic Mendelian models to more intricate polygenic and multifactorial paradigms. This shift in focus reflects the growing recognition of the complexity of NSCL/P etiology and the need to account for the contribution of multiple genes and their interactions with other factors. In this review, we aim to provide a detailed summary of the genetic inheritance patterns of non-syndromic cleft lip with or without palate (NSCL/P) within the Chinese population. We conducted a systematic search of the literature using the PubMed database (https://pubmed.ncbi.nlm.nih.gov/, accessed on 15 April 2023) up to July 2023 to ensure the inclusion of relevant and high-quality studies. Our search terms and criteria focused on NSCL/P, inheritance patterns, Chinese population, and polygenic risk scores. The selection process involved a thorough examination of titles, abstracts, and full-text articles, with a focus on peer-reviewed publications and their relevance to our research objectives.

Note: In this study, all figures were prepared using Adobe Illustrator 2023 (Adobe Inc., San Jose, CA, USA).

## 2. Genetic Etiology of CL/P

The processes of tissue fusion during the development of the upper lip and palate, including cell proliferation and migration, are regulated by multiple cellular signaling pathways, such as the Wingless-related integration site (WNT), Sonic Hedgehog (SHH), bone morphogenetic protein (BMP), transforming growth factor β (TGF-β), and fibroblast growth factor (FGF) signaling pathways [5,11,12]. Disruptions in these pathways can impede normal embryogenesis, resulting in facial dysmorphology. Furthermore, the etiology of OFCs can be intricate owing to the interplay between genetic and other contributing factors.

In SCL/P, cleft lip and cleft palate usually coincide with other systems’ abnormalities and adhere to the classic Mendelian inheritance pattern. It can be etiologically attributed to rare genetic alterations with a large effect size (Figure 2), such as single nucleotide variants (SNVs), copy number variants (CNVs), or chromosomal abnormalities. These variants may exert a substantial impact on the development of CL/P. The disease-associated genes include *IRF6*, *NECTIN1*, *CDH1*, *MSX1*, etc. For instance, Van der Woude syndrome (VWS, OMIM#119300), a dominant-inherited developmental disorder characterized by frequent occurrences of upper lip and/or palate clefts, constitutes the most prevalent orofacial cleft syndrome and accounts for 2% of all CL/P cases [13]. The heterozygous pathogenic mutation of the *IRF6* gene has been proven to be associated with numerous VWS cases [14,15]. Additionally, it has been reported by Du et al. that mutations in the *MSX1* gene are associated with congenital tooth loss in VWS patients [16]. Cleft lip/palate-ectodermal dysplasia syndrome (CLPED1, OMIM#225060) is caused by the homozygous pathogenic mutation in the *NECTIN1* gene, with clinical features including cleft lip and palate, ectodermal dysplasia, and mental retardation [17,18]. Blepharocheilodontic syndrome (BCDS1, OMIM#119580) is a rare genetic condition with specific features such as lower eyelid ectropion, wide eyelids, bilateral cleft lip and palate, as well as conical teeth [19]. This syndrome follows an autosomal dominant inheritance pattern and is caused by heterozygous pathogenic mutation of the *CDH1* gene [20,21]. The *MSX1* gene is mandatory for tooth agenesis and craniofacial development [22,23]. The abnormality of *MSX1* could result in Wolf–Hirshhorn syndrome (WHS) [24]. The corresponding phenotypes involve hypodontia, cleft lip with or without palate, and developmental delay. More genes associated with SCL/P are summarized in detail in Table 1. 

**Table 1 genes-14-01859-t001:** **Genes implicated in syndromic cleft lip with or without palate (SCL/P)**.

Gene	Location	Protein Function	Phenotype	Phenotype MIM Number ^†^	Inheritance
*ACTB*	7p22.1	β Actin	Baraitser–Winter syndrome 1	243310	AD
*CDH1*	16q22.1	Cadherin 1	Blepharocheilodontic syndrome 1	119580	AD
*EFNB1*	Xq13.1	Ephrin B1 receptor protein-tyrosine kinase	Craniofrontonasal dysplasia	304110	XLD
*ESCO2*	8p21.1	Chromatid cohesion N-acetyltransferase 2	Juberg–Hayward syndrome	216100	AR
			Roberts-SC phocomelia syndrome	268300	AR
*FGF8*	10q24.32	Fibroblast Growth Factor 8	Hypogonadotropic hypogonadism 6 with or without anosmia	612702	AD
*GLI2*	2q14.2	GLI family zinc finger 2	Holoprosencephaly 9	610829	AD
*GLI3*	7p14.1	GLI family zinc finger 3	Pallister–Hall syndrome	146510	AD
*HYLS1*	11q24.2	HYLS1 Centriolar and Ciliogenesis Associated	Hydrolethalus syndrome	236680	AR
*IRF6*	1q32.2	Interferon regulatory 6 transcription factor	Van der Woude syndrome 1	119300	AD
*KDM6A*	Xp11.3	Lysine demethylase 6A	Kabuki syndrome 2	300867	XLD
*MID1*	Xp22.2	Midline 1	Opitz GBBB syndrome	300000	XLR
*MSX1*	4p16.2	Msh homeobox 1	Wolf–Hirschhorn syndrome	194190	Unknown [24,26]
*NECTIN1*	11q23.3	Nectin cell adhesion molecule 1	Cleft lip/palate-ectodermal dysplasia syndrome	225060	AR
*OFD1*	Xp22.2	Centriole and centriolar satellite protein	Orofaciodigital syndrome I	311200	XLD
*PHF8*	Xp11.22	PHD finger protein 8	Intellectual developmental disorder, X-linked syndromic, Siderius type	300263	XLR
*RIPK4*	21q22.3	Receptor interacting serine/threonine kinase 4	Popliteal pterygium syndrome, Bartsocas–Papas type 1	263650	AR
*TFAP2A*	6p24.3	Transcription factor AP-2 α	Branchiooculofacial syndrome	113620	AD
*TP63*	3q28	Tumor protein p63	Ectrodactyly, ectodermal dysplasia, and cleft lip/palate syndrome 3	604292	AD
			Hay–Wells syndrome	106260	AD
			Rapp–Hodgkin syndrome	129400	AD
*WNT3*	17q21.31-q21.32	Wnt family member 3	Tetra-amelia syndrome 1	273395	AR

^†^ Phenotype identifier number in Online Mendelian Inheritance in Man (OMIM^®^) database (https://omim.org, accessed on 30 April 2023). Abbreviations: AD, autosomal dominant; AR autosomal recessive; XLD, X-linked dominant; XLR, X-linked recessive.

In comparison to SCL/P, NSCL/P exhibits a higher disease incidence but lower genetic interpretability. To date, only a few studies have elucidated the causal genes/mutations of NSCL/P. For example, several pathogenic/likely pathogenic variants have been identified and validated in some genes (e.g., *CTNND1*, *PLEKHA5*, and *ESRP2*) that influence the expression of epithelial Cadherin-p120-Catenin complex in multi-affected NSCL/P families [27]. Population-based and family-based analyses identified that *PAX9* and *TGFB3* might contribute to NSCL/P as well [28,29]. However, according to previous twin studies, 40–60% of monozygotic twins shared the concurrence of NSCL/P and similar traits, while only 3–5% of dizygotic twins displayed such concurrence, suggesting a strong tendency of genetic inheritance [30]. Therefore, it is suggested that the isolated phenotype of NSCL/P is attributed to the cumulative impact of multiple common genetic alterations with modest effect sizes, such as single nucleotide polymorphisms (SNPs) (Figure 2).

Previous genetic investigations of NSCL/P included direct panel sequencing or linkage analysis in large pedigrees with family histories or small pedigrees involving consanguineous marriages [7]. The advent of next-generation sequencing (NGS) has facilitated high-throughput genetic analysis via whole-exome sequencing (WES) and whole-genome sequencing (WGS) in extensive cohorts. In contrast to panel testing, which sequences established candidate loci, WES/WGS can aid in uncovering novel pathogenic regions and leverage multiple candidate relative variants for statistical risk evaluation. Nonetheless, the scarce accessibility of genomic tools and the demand for more expansive datasets continue to hinder progress in comprehending the genetics of NSCL/P. Therefore, completing the genetic architecture of NSCL/P inheritance remains an ongoing challenge. As NSCL/P represents a multifactorial condition with a wide phenotypic spectrum and a limited number of causal genes directly accountable for the disorder, it is presumed not to conform to the conventional Mendelian inheritance model [31]. In this review, we will explore the genetic inheritance models of NSCL/P from a polygenic inheritance perspective, with a particular emphasis on the Chinese population, which has yet to be examined comprehensively.

## 3. Genome-Wide Association Study (GWAS) of NSCL/P Worldwide

The genome-wide association study (GWAS) has rapidly evolved as an advancing technique for identifying single-nucleotide polymorphism loci exhibiting significant disparities in the sequencing results between case and control cohorts. This method is well-suited for analyzing complex traits, wherein multiple loci, rather than a single rare variant, are more likely to serve as additive contributing factors. Ludwig et al. performed the first GWAS of NSCL/P in 2012, validating previously recognized loci related to NSCL/P and uncovering six additional regions with susceptibility (1p36, 2p21, 3p11.1, 8q21.3, 13q31.1, and 15q22) [32]. Among the specific genetic risk factors, rs8001641 was initially reported and subsequently replicated in other studies investigating the genetic association [33,34,35]. Since then, dozens of genes and significant regions that could be retrieved from the GWAS catalog database have been implicated in NSCL/P [36]. A comprehensive summary of these findings up to 2020 has been provided by Nasreddine et al. [2].

One of the pitfalls of GWAS is that the result is easily confounded by different ancestries of selected samples, case–control imbalance, various sequencing methods, and linkage disequilibrium, leading to poor stability and transferability across different study groups [37,38,39]. However, with the progression of GWAS analysis tools, these confounding factors could be minimized to a lower level. For example, SAIGE, developed by Zhou et al. [40], employs saddle-point approximation (SPA) to address the unbalanced case–control ratio issue and offers an algorithm to construct a logistic/linear mixed model to compute sample relatedness. Another example for solving the ancestry problem is Tractor [41], which estimates specific effect sizes according to population structure and thus boosts GWAS analysis power. Leveraging these analysis tools, cohort analysis is able to unveil more novel and biologically relevant SNPs/indels of NSCL/P. 

## 4. Association Studies and Related Loci of NSCL/P in the Chinese Population

Among diverse ethnic groups, the significant loci identified by GWAS may vary considerably due to diverse geographic and ancestral population structures. For now, the majority of GWASs are Eurocentric, which introduces bias when analyzing other ancestries [42]. Therefore, in addition to developing novel statistical methods to enhance stability and transferability, identifying geographic-specific significant loci is of great importance. In the following sections, we will discuss several well-established genes and their loci with strong relevance to NSCL/P among the Chinese population. A comprehensive summary of significant loci related to NSCL/P that were reported in the Chinese population is shown in Table 2. 

**Table 2 genes-14-01859-t002:** **Significant loci associated with non-syndromic cleft lip with or without palate (NSCL/P) in Chinese population**.

SNP ID	Affected Allele	Gene/Region	OR (95% CI)	*p*-Value	Population	Method	Case: Control	Year	PMID
rs7650466	T	*EPHA3*	0.211(0.131–0.338)	4.88 × 10^−10^	Han Chinese	Targeted sequencing	180:167	2018	29932736 [43]
rs58593329	A	*VAX1*	1.34 (1.2–1.5)	1.90 × 10^−7^	Western Han Chinese	Targeted sequencing	1626:2255	2022	35419918 [44]
rs11197887	A		1.35 (1.21–1.51)	8.52 × 10^−8^					
rs1904302	T		1.39 (1.24–1.54)	2.66 × 10^−9^					
rs10886040	G		1.40 (1.26–1.56	9.50 × 10^−10^					
rs744937	T		1.39 (1.25–1.54)	2.23 × 10^−9^					
rs7078160	A		1.40 (1.25–1.55)	1.12 × 10^−9^					
rs17095681	T	*SHTN1*	0.64 (NA)	3.80 × 10^−9^	Han and Hui Chinese	GWAS and Targeted sequencing	1931:2258	2016	28008912 [45]
rs4791331	T	*NTN1*	1.43 (1.20–1.70)	5.10 × 10^−5^	Han Chinese	Targeted sequencing	873:830	2020	31780810 [46]
rs2235371	T	1q32.2	0.67 (0.62–0.73)	8.69 × 10^−22^	Chinese	GWAS	858:1248	2015	25775280 [47]
rs7078160	A	10q25.3	1.29 (1.19–1.39)	3.09 × 10^−10^					
rs8049367	T	16p13.3	0.74 (0.68–0.80)	8.98 × 10^−12^					
rs4791774	G	17p13.1	1.56 (1.42–1.72)	5.05 × 10^−19^					
rs13041247	C	20q12	0.76 (0.71–0.83)	1.69 × 10^−11^					
rs17820943	T	20q12	-	6.70 × 10^−5^	Southern Han Chinese	Targeted sequencing	430:451	2020	31713353 [48]
rs6072081	G		-	4.52 × 10^−4^					
rs6072081	G	20q12	0.72 (0.58–0.9)	4.00 × 10^−3^	Han Chinese	Targeted sequencing	305:356	2012	22522387 [49]
rs13041247	C		0.68 (0.54–0.85)	7.20 × 10^−4^					
rs6102085	A		0.62 (0.49–0.77)	2.14 × 10^−5^					

Abbreviations: OR, odds ratio; CI, confidence interval; NA, not applicable; PMID, PubMed identifier.

### 4.1. VAX1

The *VAX1* gene, located on chromosome 10q25.3, encodes a transcription factor with a highly conserved homeodomain DNA-binding motif. This motif enables these proteins to regulate the expression of downstream target genes. The protein product of VAX1 is essential in embryonic development, contributing to the formation of the eyes, nose, and upper jaw. 

Multiple lines of evidence have linked *VAX1* to cleft lip and palate development. In mouse models, loss of Vax1 expression has demonstrated abnormal craniofacial development, including cleft palate [50]. Similarly, genetic studies in humans have identified *VAX1* mutations associated with the phenotype [31,51,52]. With target sequencing of 1626 NSCL/P patients and 2255 controls in the Western Han Chinese population, You et al. replicated one previously reported *VAX1* SNP (rs7078160) and identified five additional SNPs exhibiting significant associations with NSCL/P risk, suggesting that *VAX1* may contribute to the disease development [44]. 

### 4.2. 20q12

Investigations have identified certain genetic variants within the 20q12 region associated with an elevated risk of NSCL/P in both European and Asian ancestries [53,54]. It is postulated that these variants may impact the development of lip and palate by regulating the expression of genes that involve cell proliferation, differentiation, and apoptosis during embryonic development. However, SNPs associated with NSCL/P in specific ancestries have not been fully identified yet because of different genetic backgrounds. One of the most frequently reported SNPs in 20q12, rs13041247, was initially identified to be associated with NSCL/P with a *p*-value of 0.0161 in the East Asian population and 0.0002 in the European and Euro-American population [51]. This SNP was subsequently confirmed in the Chinese Han population respectively through GWAS and targeted sequencing in large cohorts respectively, supplementing the cross-ancestry genetic architecture of NSCL/P [47,49]. Another case–control analysis involving 430 patients and 451 controls also revealed significant associations of rs17820943 and rs6072081 at 20q12 in Southern Han Chinese [48].

Given that NSCL/P is a complex condition with multiple genetic and environmental factors contributing to its development, the association between these regions and NSCL/P constitutes only a small part of a much larger puzzle. Continued research is required to further comprehend the underlying mechanisms contributing to NSCL/P and to devise more effective prevention and treatment strategies.

## 5. Polygenic Inheritance and Polygenic Risk Score in NSCL/P

GWAS yields two parameters of specific loci that can be utilized for subsequent calculations: significance (*p*-value) and effect size (β) [55]. Loci with positive effect sizes may be considered as risk factors, while those with negative effect sizes may be regarded as protective factors. Since GWAS solely interprets the association between SNP loci and disease, and most SNP loci have relatively small effect sizes, these loci are considered to be more indicative of risk, rather than the whole genetic landscape of the disease [56]. 

A rational approach is to integrate multiple genetic loci and their effects into polygenic risk scores (PRS), which more accurately reflect the genetic foundation of a phenotype or disease in different disease groups. With the accumulation of genomic data and the establishment of large prospective population-based cohorts, such as the China Kadoorie Biobank (CKB) [57] and the UK Biobank (UKB) [58], the predictive power of PRS in a variety of complex diseases and phenotypes has been independently and prospectively evaluated [59,60]. PRS has been demonstrated to function as a genetic indicator representing genetic intensity or genetic risk for population risk stratification, with potential applications in predicting disease risk, treatment selection, and disease prognosis estimation to advance precision medicine. To calculate PRS for a disease of interest, researchers can use either in-house association results (SNPs and their effect sizes) or existing published association results from the PGS catalog (https://www.pgscatalog.org/, accessed on 15 April 2023). The PGS catalog is a comprehensive database that consolidates polygenic score data from various studies, offering a centralized resource for researchers and clinicians. It provides information on the predictive power of polygenic scores for different traits and diseases, including the specific genetic variants, effect sizes, and sample populations used in each score. This valuable tool aids in understanding the genetic basis of complex traits and diseases, enabling personalized medicine approaches based on an individual’s genetic risk profile. State-of-the-art software products for calculating PRS include LDpred [61], SBayesR [62], PRS-CS [63], and PRSice-2 [64]. The utilization of PRS exhibits promising potential for precise prediction, prevention, and personalized treatment of complex diseases. Notably, PRS has been extensively employed in risk prediction and stratification for breast cancer [65,66] and cardiovascular diseases [67,68]. Nevertheless, the application of PRS in the context of NSCL/P remains limited [69,70]. Figure 3 provides a brief depiction of a plausible clinical process for application of the polygenic risk score for NSCL/P.

To the best of our knowledge, no PRS study has specifically investigated NSCL/P within the Chinese population. As sequencing cohorts expand, more SNP candidate loci with high quality and strong associations are expected to be discovered and utilized in PRS calculation. Furthermore, although data are currently limited, integrating rare and common variants may be another approach to enhance risk estimation and explain some unusual genetic patterns. For instance, Yu et al. reported that in a large NSCL/P pedigree, the penetrance of a putatively causal variant in *PDGFRA*(c.C2740T) was modified by additional common variants [71]. However, as previously mentioned, the remarkable heterogeneity of SNP heritability of different ancestries is a major obstacle to PRS calculation. Europe possesses a more complete sequencing database, and the majority of GWASs are now conducted within European populations, which results in poor transferability to other populations such as Asians. Sequencing of large samples and subsequent analysis for NSCL/P needs to be prioritized in Asian populations to explore the genetic architecture and to facilitate risk prediction and stratification for NSCL/P.

## 6. Conclusions and Perspectives

The identification of environmental factors and the investigation of gene–environment interactions requires not only extensive cohort studies but also access to genetic material for optimal results. During early embryonic development, certain environmental factors can induce abnormalities through a variety of mechanisms, resulting in cleft lip and/or palate. It has been demonstrated that maternal active/passive smoking during the first pregnancy trimester could significantly elevate the risk of NSCL/P [72]. Furthermore, alcohol consumption [73], drug use [74], increased maternal age at birth [75], infections during pregnancy, and history of miscarriage also contribute to risk [76]. The intake of folic acid and vitamins is another crucial factor. Studies have revealed that 5,10-methylenetetrahydrofolate reductase (MTHFR) is essential in lip and palate development and mandatory folic acid fortification is effective in reducing the risk [77]. High temperature, stress, maternal obesity, occupational exposure, and ionizing radiation have also been associated with NSCL/P [76,78,79]. However, consensus regarding the detrimental impacts of these factors is lacking, and large-scale retrospective and prospective cohort studies may be required. 

Phenotyping is one of the most important steps in clinical genetic analysis. Accurate phenotyping provides valuable references for data curation. NSCL/P is a heterogeneous disease with complex traits that can be categorized into various subtypes. In patients with milder phenotypes, subtle subclinical phenotypes may go unnoticed. Detailed clinical information should be meticulously documented and can be supplemented with advanced measuring machines, e.g., high-resolution or 3D prenatal sonography.

It is worth mentioning that both GWAS and PRS focus on the gene level. For a more in-depth exploration at the expression level, RNA sequencing is required. Integrative analysis of expression quantitative trait loci (eQTL) and GWAS results can shed light on how the variants affect gene expression and the function of their products. Since NSCL/P is a multifactorial disease, dissecting its etiology from both genetics and environment would be more comprehensive. The utilization of the genetics*environment (G*E) model makes it possible to take environmental factors into consideration. The G*E interaction model not only calculates the effects of specific environmental or genetic risk factors but combines them in a statistically explainable manner [80]. Nevertheless, such models are not yet easily accessible for widespread use. The primary reason is that environmental factors, unlike genetics, cannot be as tightly controlled and can vary over time. Additionally, environmental factors can be both binary and continuous variables, and cross-validation may require significant computational resources, making it challenging to incorporate them into a mathematical model in a comprehensible and reliable manner.

It is essential to acknowledge the limitations of this study. Firstly, the focus of this review was primarily on the Chinese population, and while it offers some insights, it may not fully represent the genetic diversity seen in other populations. Additionally, here, we only chose some representative loci/genes/regions to elucidate the inheritance pattern of NSCL/P. The formation of the lip and cleft represents fundamental embryogenic processes intricately governed by numerous signaling pathways. While we have provided a summary of several genes and loci associated with CL/P, it is imperative to underscore the necessity of considering additional related loci, genes, and regions in a more comprehensive investigation. Finally, while we discuss the potential of polygenic risk scores for risk stratification, it is essential to note that their practical clinical application and validation require further research. Several key considerations merit attention, including the availability of suitable GWAS summary statistics for the target population and the judicious selection of computational methods to maximize predictive accuracy.

In conclusion, our review delves into the intricate genetic factors underlying non-syndromic cleft lip with or without palate (NSCL/P) in the Chinese population. We explained the genetic inheritance models from monogenic to polygenic perspectives, presented representative genes and regions, and highlighted the potential of polygenic risk scores for personalized risk assessment. Multiple genes and their loci are associated with the development of NSCL/P worldwide, including the Chinese population. However, experimental data findings could be inconsistent across populations and sometimes even contradictory, possibly due to the intricate ethnic composition of the Chinese population and the genetic heterogeneity of NSCL/P. Furthermore, the occurrence of such a complex disease is also influenced by environmental factors. As research methods and techniques advance, it is imperative to further explore the genetic background and underlying mechanisms by integrating multiple genetic analysis strategies. Elaborate phenotyping with polygenic risk scores could be utilized for calculating NSCL/P to normal facial variation and to investigate genes with known effects on facial morphology under certain conditions. The combination of genetic and environmental risks with gene expression, system biology, epigenetics, and epidemiology promises to yield a more comprehensive etiological profile, providing new avenues for early disease screening, diagnosis, and prevention of cleft lip and palate, as well as better clinical care and prevention.

## Figures and Tables

**Figure 1 genes-14-01859-f001:**
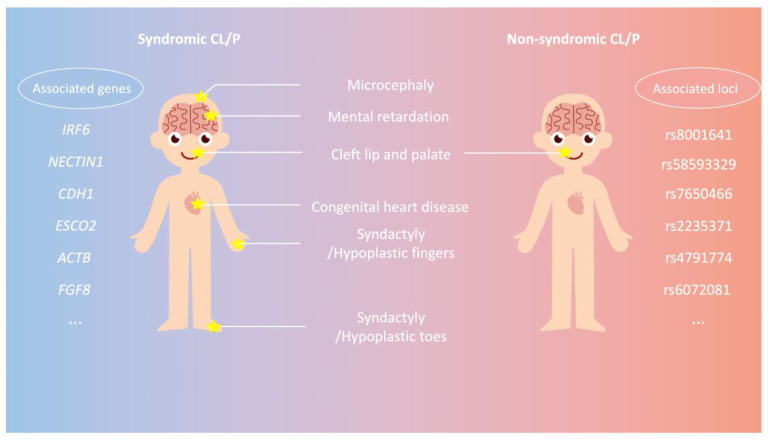
Phenotypes and corresponding genes/loci associated with syndromic and non-syndromic cleft lip with or without palate (CL/P). Syndromic CL/P individuals (left panel) typically exhibit multisystem involvement, affecting head, brain, mouth, limbs, etc. Several genes identified as associated with syndromic CL/P include *IRF6*, *NECTIN1*, *CDH1*, *ESCO2*, *ACTB*, and *FGF8* (refer to Table 1 for details). In contrast, non-syndromic CL/P individuals (right panel) primarily manifest the phenotype of CL/P without any additional abnormalities in other systems. The loci linked to non-syndromic CL/P comprise rs8001641, rs58593329, rs7650466, rs2235371, rs4791774, and rs6072081 (see Table 2 for details).

**Figure 2 genes-14-01859-f002:**
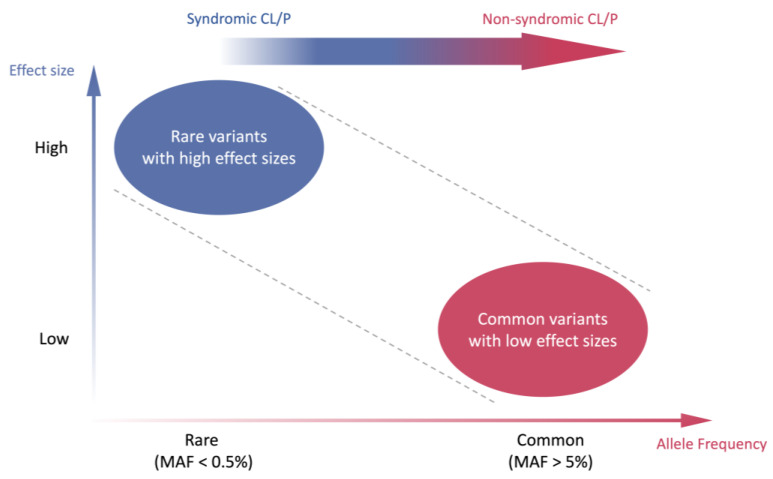
**Postulated genetic etiology of syndromic and non-syndromic cleft lip with or without palate (CL/P) by risk allele frequency and effect size.** Variants are categorized into two groups by risk allele frequency (x-axis) and effect size (y-axis). Rare variants with high effect sizes are depicted in the upper left quadrant as a blue circle, while common variants with low effect sizes are represented in the lower right quadrant as a red circle. Syndromic CL/P, which follows a classic Mendelian inheritance pattern, is more likely to be attributed to rare variants with high effect sizes (blue circle). In contrast, non-syndromic CL/P, which displays a genetic inheritance tendency without clear identification of responsible variants, is proposed to be influenced by the cumulative effects of multiple common genetic alterations with low effect size (red circle). CL/P, cleft lip with or without palate; MAF, minor allele frequency. This figure is adapted from McCarthy et al., 2008 [25].

**Figure 3 genes-14-01859-f003:**
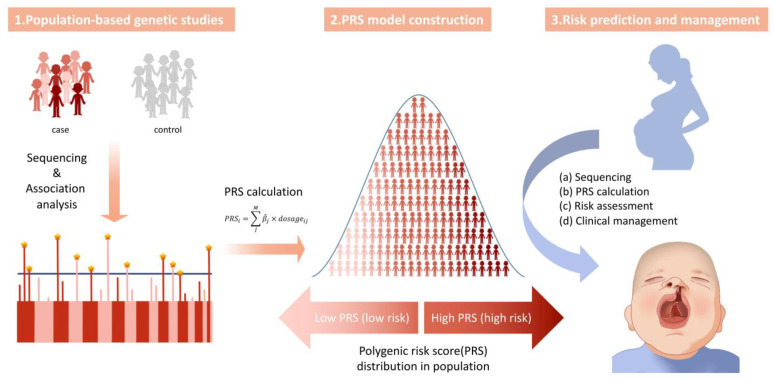
**Clinical process for application of polygenic risk score (PRS) for non-syndromic cleft lip with or without palate (NSCL/P).** 1. left panel: Affected individuals and unrelated healthy controls undergo sequencing, and the results are utilized in association studies to identify genetic signals and calculate effect sizes. The lower left section is a simplified Manhattan plot, which serves as a commonly used tool for interpreting genome-wide association study (GWAS) analysis results. 2. middle panel: Using variant information from the association studies, PRS is calculated with the formula ${\mathrm{PRS}}_{i}=\overset{M}{\underset{j}{\sum }}\beta_{j}\times {\mathrm{dosage}}_{\mathrm{ij}}$, which involves a weighted sum of effect sizes and dosages (0/1/2) of all variants. The calculation is performed on each person of a separate group of individuals different from the population used in generating the association study results. The PRS values are then plotted into a normal distribution graph, assigning each individual a position indicating their relative risk for NSCL/P within the population. 3. right panel: In a clinical scenario where a pregnant woman desires to ascertain her baby’s risk of developing NSCL/P, the following steps are outlined: (a) Genetic sequencing is conducted on the baby; (b) the baby’s PRS is calculated using the aforementioned formula; (c) by comparing the baby’s PRS to the population PRS distribution plot, the relative risk of the baby (risk percentile) can be determined; (d) the PRS, along with other prenatal examinations such as ultrasound, assists clinicians in making informed clinical decisions.

## Data Availability

No new data were created or analyzed in this study. Data sharing is not applicable to this article.

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
