# Peer review of "Genetic Inheritance Models of Non-Syndromic Cleft Lip with or without Palate: From Monogenic to Polygenic"

_genes, 2023, doi:10.3390/genes14101859_

Round 1
Reviewer 1 Report
Dear Authors,
thank you for the article. Here are some suggestions on how to improve it:
1. In the abstract, you write about 1:700 prevalence. Than in the introduction, you write about 1:800. Change that _+ in the introduction add more information on sex prevalence and racial prevalence.
2. When describing the process of creating lip and palate, please write about gestation.
3. All the abbreviations should be explained when firstlu used. Due to the fact, that there are plenty of them (including the genes nomenclature), please use a table at the end (or as supplementary material) to sum them up.
4. In the introduction, you should write more about the speciffic genes showing some syndromes, eg. MSX1 -vs. Wolf-Hirshhorn syndrome etc. You should also highlight the most frequent genes in the etiology of cleft (MSX1, PAX9, TGF-B, IRF-6 etc), eg.
- Nasroen SL, Maskoen AM, Soedjana H, Hilmanto D, Gani BA. IRF6 rs2235371 as a risk factor for non-syndromic cleft palate only among the Deutero-Malay race in Indonesia and its effect on the IRF6 mRNA expression level. Dent Med Probl. 2022;59(1):59–65. doi:10.17219/dmp/142760
5. The paragraph 2 misses the most common genes responsible for cleft occurance
6. Line 54 - please, give the deffinition of syndromic and non-syndromic CLP, as it may not be clear
7. Line 58, please add gene responsible for VDW syndrome
8. Line 175-178 should not base on one reference only (50), you should are more references regarding multifactorial and environmental influence on inheritance pattern.
9. Line 177 - it refers to the supplementation of high doses of folic acid (not "typical" ones) and it should be stated here.
10. Line 196 - I would change "and within Chinese..." to "including Chinese"
11. More genes are involved in the occurence of cleft and the research should be widened to them, or the title should be changed to suggest that those are "chosen genes" and it should be clear throughout the whole paper - you should highlighen that
12. Are the figures originally prepared for the paper - are they prepared by Authors?
13. The limitations of the study is missing. It should sum up the discussion section.
14. The conclusions are mandatory
15. Due to the fact, that most of the paper refers to Chinese population, it should be stated in the title.
Thank you.
Author Response
Thank you so much for the critical assessment and constructive comments on our manuscript discussing the inheritance pattern of non-syndromic cleft lip with or without palate. Please kindly find attached the response to the comments.

Reviewer 2 Report
1. What type of inheritance pattern is cleft lip and palate?
2. What is the difference between Nonsyndromic and syndromic cleft palate?
3. Genetic markers for non-syndromic orofacial clefts?
4. What are the Signaling pathways involved?
Author Response

(The authors gave the same response as above.)

Reviewer 3 Report
The article “Genetic inheritance models of non-syndromic cleft lip with or without palate: from monogenic to polygenic” is interesting and I have some comments to make to improve the article.
- Add at the end of the “background” how you selected the articles to perform the review
- Where does the reference “with a prevalence of approximately 1.7/1000 worldwide” come from (line 67)?
- The following paragraph must be accompanied by a reference “GWAS yields two parameters of specific loci that can be used for subsequent calculations: significance (p-value) and effect size (beta). Loci with positive effect sizes may be considered as risk factors, while those with negative effect sizes may be considered as protective factors. Since GWAS solely interprets the association between SNP loci and disease, and most SNP loci have relatively small effect sizes, these loci are considered to be more indicative of risk, rather than the whole genetic landscape of the disease.”
- Tables and figures. Add bibliographic references from where the information is collected. - Add the reference of McCarthy et al., 2008.
Thank you
Author Response

(The authors gave the same response as above.)

Round 2
Reviewer 1 Report
Thank you for this corrections done to your paper. There are some more suggestions to be solved. Beside that, the paper really improved:
1. VDW syndrome is often associated with MSX1 gene, please add that.
2. If the figure was taken from previous study and had been published already, you should mention that in the text and refer to the prevalence of copying it.
3. In the text, please state who prepared the figures and which program was used to that.
Besides, please add more recent references as they should be presented in the genetic paper - this is very much developing branch, therefore it should contain papers from years 2013 and up.
Author Response
Thank you so much for the further constructive comments on our manuscript! Please kindly find attached the response to the comments.
